# Global divergence in plant and mycorrhizal fungal diversity hotspots

Laura G. van Galen [1,2] ✉, Justin D. Stewart[1,3], Clara Qin[1], Adriana Corrales [1], Bethan F. Manley [1], E. Toby Kiers[1,3], Thomas W. Crowther [2] & Michael E. Van Nuland[1]

Environmental protection strategies often rely on aboveground biodiversity indicators for prioritising conservation efforts. However, substantial biodiversity exists belowground, and it remains unclear whether aboveground diversity hotspots are indicative of high soil biodiversity. Using geospatial layers of vascular plant, arbuscular mycorrhizal fungi, and ectomycorrhizal fungi alpha diversity, we map plant–fungal diversity associations across different scales and evaluate evidence for potential correlation drivers. Plant–fungal diversity correlations are weak at the global scale but stronger at regional scales. Plant–arbuscular mycorrhizal fungal correlations are generally negative in forest biomes and positive in grassland biomes, whereas plant–ectomycorrhizal fungal correlations are mostly positive or neutral. We find evidence that symbiosis strength, environmental covariation, and legacy effects all influence correlation patterns. Only 8.8% of arbuscular mycorrhizal and 1.5% of ectomycorrhizal fungal diversity hotspots overlap with plant hotspots, indicating that prioritising conservation based solely on aboveground diversity may fail to capture diverse belowground regions.

Protecting biodiversity is critical for maintaining ecosystem health and resilience in the face of global change[1,2]. Despite being the most diverse community on land[3] and key drivers of almost all ecosystem processes[2,4,5], soil organisms are rarely considered in conservation planning[6–8]. Conservation biases towards aboveground macroorganisms may be problematic when the patterns of belowground biodiversity diverge strongly from those aboveground[9–13]. We currently lack a comprehensive global overview of the directions and strengths of relationships between plant and fungal diversity, and how they vary across scales[14]. For example, biological relationships may differ depending on whether interactions are examined at the global level or at smaller biome and ecoregion (finer-scale climatic divisions within biomes[15]) scales[10]. Additionally, the degree to which the 'hotspots' of plant diversity coincide with those of fungal diversity remains unknown. This information is critical for understanding aboveground–belowground diversity linkages and

designing conservation strategies that adequately capture the vast diversity present belowground.

Plant–mycorrhizal fungal diversity relationships are of particular interest, because mycorrhizal symbioses occur in over 90% of vascular plant species and have substantial impacts on ecosystem productivity and function[16]. Yet, a range of interacting factors are likely to shape the diversity patterns of these two groups. It has been proposed that mutualisms might lead to the 'coupling' of diversity, where the diversity of one group leads to direct increases in the diversity of the other by expanding resource availability and niche space[5,14,17,18]. In such situations, positive diversity correlations between vascular plants and mycorrhizal fungi are likely to be stronger when potential host plant species are more prevalent and comprise a larger proportion of the plant community (i.e. when plant diversity is a stronger proxy for host diversity; Fig. 1A, B). However, diversity correlations can also arise independently of diversity coupling. If plants and microbes have

[1]Society for the Protection of Underground Networks (SPUN), Dover, DE, USA. [2]Department of Environmental Systems Science, Institute of Integrative Biology, ETH Zürich (Swiss Federal Institute of Technology), Zürich, Switzerland. [3]Amsterdam Institute for Life and Environment (A-LIFE) Section, Ecology & Evolution, Vrije Universiteit Amsterdam, Amsterdam, The Netherlands. ✉e-mail: laura.vangalen9@gmail.com

## Symbiosis effects

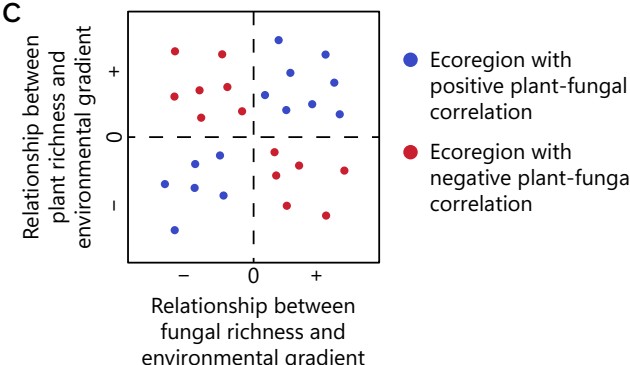

## Environmental effects

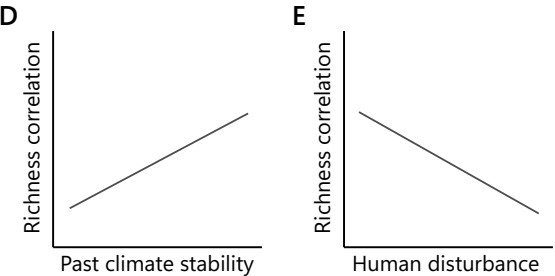

## Legacy effects

**Fig. 1 | The potential mechanisms driving plant–fungal richness correlation patterns.** If symbiosis-driven diversity 'coupling' is influencing diversity relationships we would expect **A** richness relationships to be stronger between arbuscular mycorrhizal (AM) fungi and plants compared to ectomycorrhizal (ECM) fungi and plants due to the majority of vascular plants (72%) being AM mycorrhizal host species and only 2% being ECM hosts[16]. We might also expect **B** richness correlations to be stronger and more positive in regions where more of the plant species or vegetation biomass belong to potential host plants (i.e. such that all-plant diversity is a stronger proxy for host plant diversity). If richness correlations are strongly influenced by how species respond to external environmental gradients, we would expect that **C** more positive correlations would arise in regions where plant and fungal richness respond similarly to an environmental gradient (top right and bottom left segments), and negative correlations to arise where they respond differently (top left and bottom right segments). If legacy effects originating over either long-term evolutionary time scales or from more recent disturbance history influence correlation patterns we might expect that **D** richness correlations are stronger and more positive in regions where past climate fluctuations have been more stable (due to evolution of plants and fungi occurring at more consistent rates under stable climates), and that **E** richness correlations become weaker or negative with increasing human disturbance (particularly for AM plant–fungal correlations if disturbance increases AM fungal diversity but reduces plant diversity[25,26]).

different environmental tolerances, positive relationships might form when plants and fungi respond similarly to the same environmental gradient, and negative correlations could form when they respond differently[19,20] (Fig. 1C). Legacy effects from past diversification

patterns, or ecological or anthropogenic disturbances, have also been shown to influence diversity relationships[21–23]. Past climate fluctuations could disrupt host–symbiont co-evolution and lead to a weakening of diversity correlations[14,24] (Fig. 1D). Additionally, disturbances affect plants and fungi in different ways[25,26] which could lead to weakened or negative diversity correlations (Fig. 1E). However, despite theoretical and experimental support for these hypotheses (symbiosis effects, environmental effects, and legacy effects) in influencing diversity associations, evidence for these drivers remains unexplored at the global level[14].

Here, we use recently-developed high-resolution geospatial layers of alpha diversity estimates for vascular plants[27,28], arbuscular mycorrhizal (AM) fungi and ectomycorrhizal (ECM) fungi[26,29] to evaluate the strength and direction of plant–fungal diversity relationships across different spatial scales. Specifically, we: (1) test if plant–fungal richness correlation strengths vary at global, biome and ecoregion scales by mapping the spatial distribution of correlations, (2) evaluate evidence regarding the contribution of symbiotic effects, environmental effects, and legacy effects in structuring correlation patterns, and (3) calculate the extent of overlap between plant and fungal diversity hotspots. This will further our understanding of the mechanisms driving diversity relationships, and test where plant diversity can be used as an adequate proxy for mycorrhizal fungal diversity and where the conservation of these groups may need to be considered independently.

## Results

### Richness correlations across scales

We first asked if plant–fungal correlation strengths varied at global, biome and ecoregion scales (as defined by Olson, et al.[15]) by mapping the spatial distribution of Spearman rank correlation coefficients from randomly selected grid cells from the alpha diversity geospatial layers[26–29] (alpha diversity predictions from different geospatial layers were combined for each taxa to form consensus richness maps; see 'Methods'). At the global scale, the diversity of both mycorrhizal types correlated positively with plant diversity, but the strength of this relationship was moderately weak for AM ($r = 0.4$) and even weaker for ECM ($r = 0.04$; Fig. 2A, D). Diversity correlations were stronger, and both positive and negative, at smaller spatial scales (biome scale: from $r = -0.48$–0.47 for AM and −0.12–0.44 for ECM; ecoregion scale: $r = -0.76$–0.89 for AM and −0.76–0.84 for ECM; Fig. 2). Interestingly, the distribution of these positive and negative associations showed clear biogeographic trends across different biomes (Fig. 2C, F). Specifically, plant–AM fungal correlations calculated within ecoregions were overall significantly negative in tropical moist forests and temperate broadleaf forests, and positive in tropical grasslands, temperate grasslands, montane grasslands, boreal forests, tundra, and deserts (one-sample $t$-tests; Supporting Information Fig. S1 and Table S1). Plant–ECM fungal ecoregion correlations were significantly positive in all biomes except for tropical coniferous forests (where correlations were not significantly different from zero; Fig. S1 and Table S1).

### Potential drivers of richness correlations

We evaluated evidence for potential drivers of the raw plant–fungal richness correlation patterns (Fig. 2) by testing the hypotheses described in Fig. 1. If symbiosis-driven effects are shaping the correlations between plant and fungal richness, we expected that plants and AM fungal richness would be more strongly related than plants and ECM fungal richness (Fig. 1A). Indeed, richness correlations between AM fungi and plants were stronger and more positive than for ECM fungi at the global level. This was found in both the raw plant–fungal richness correlations (Fig. 2A, D), and after removing variation explained by environmental covariates (see 'Methods'; Fig. 3A). However, contrary to our hypothesis that higher prevalence of mycorrhizal host plants would lead to stronger diversity coupling (Fig. 1B), we did not observe stronger positive correlations in ecoregions containing a

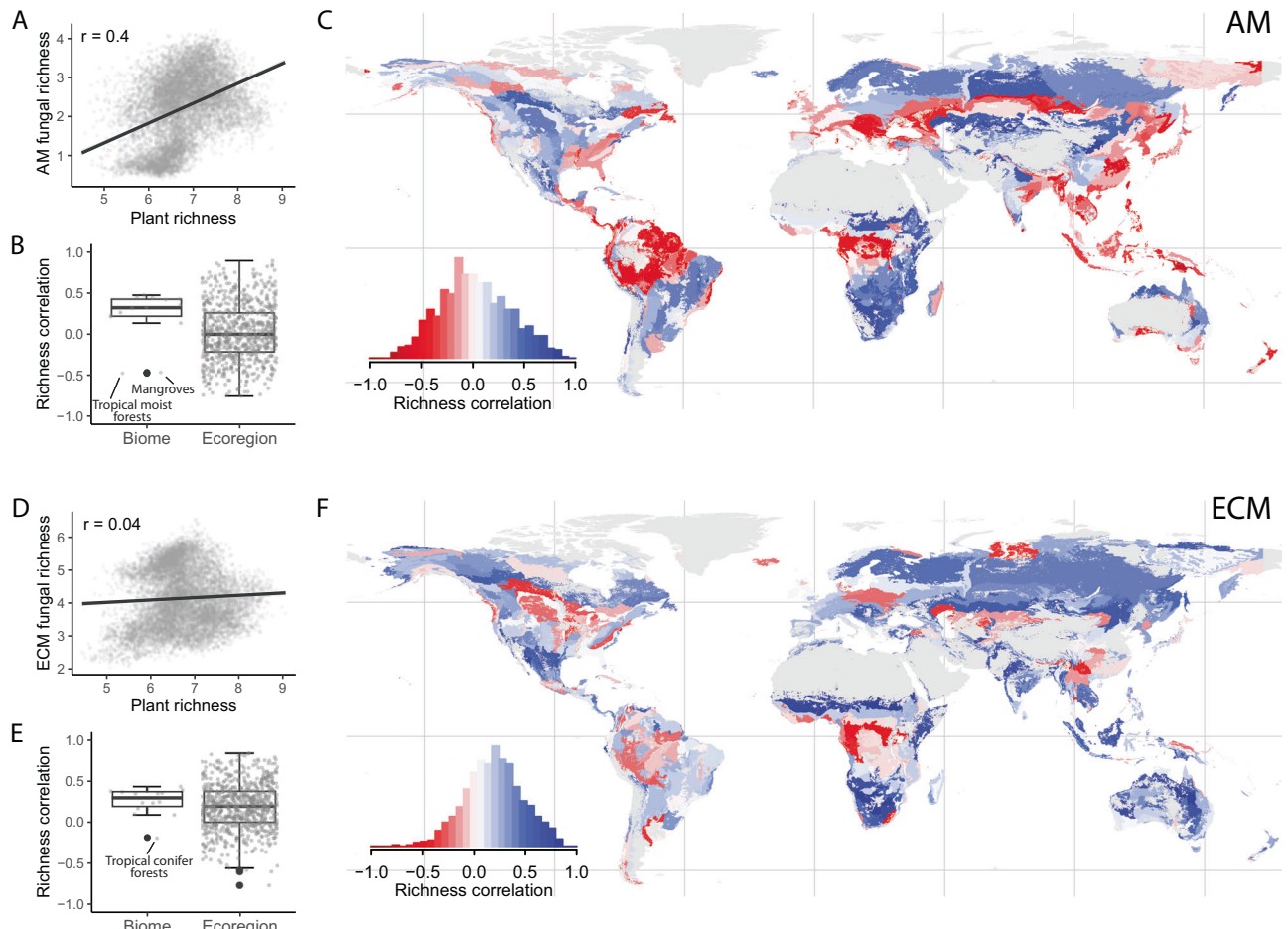

**Fig. 2 | Plant–fungal richness correlations (Spearman) at the global, biome, and ecoregion levels.** Correlations are shown for arbuscular mycorrhizal (AM) fungi (**A**–**C**) and ectomycorrhizal (ECM) fungi (**D**–**F**). **A**, **D** Show global-level relationships based on values from 10,000 randomly selected grid cells of the consensus plant and fungal richness maps (see 'Methods'). Trend lines show linear fits. **B**, **E** Show the distribution of correlations re-calculated within biomes and ecoregions (from 10,000 randomly selected grid cells in each biome and 1000 cells in each ecoregion; boxes show median and interquartile range, whiskers show 1.5 times the interquartile range). Maps (**C**, **F**) show ecoregion polygons[15] coloured by correlation strength. Negative correlations are shown in red and positive correlations in blue. Biome correlation outliers in (**B**, **E**) are labelled with biome names. Grey areas were excluded from the analyses due to high uncertainty in the original alpha diversity predictions or where predictions for each taxonomic group by the different studies strongly disagreed (mostly based on the coefficient of variation between bootstrap model replicates; see 'Methods'). Maps showing non-linear relationships from GAMs between plant and fungal richness at the ecoregion scale, and a comparison between the Spearman correlation coefficients and deviance explained by the GAMs, are provided in Supporting Information Figs. S2 and S3.

greater percentage of vegetation biomass belonging to host plant species (Fig. 3B).

If environmental covariates are shaping richness correlations, we hypothesised that the plant–fungal richness correlation direction (positive or negative) would be related to whether plant and fungal richness were responding similarly or differently to environmental gradients (Fig. 1C). Both plant–AM fungal and plant–ECM fungal correlations were more often positive when plant and fungal richness responded to temperature in the same direction, and more often negative when they responded in different directions (Fig. 3C). This pattern was also evident for precipitation and soil pH, but only for plant–AM correlations (Fig. 3C). Environmental covariation may also explain the prevalence of negative plant–fungal diversity associations observed between AM fungi and plants in some biomes. Summarising Fig. 3C by biomes (Supporting Information Fig. S4) showed that, in both tropical moist forests and temperate broadleaf forests, AM fungal richness was positively related to soil pH levels (i.e. higher diversity in more basic soils), whereas plant richness was negatively related. Therefore, soil pH may be an important driver of the negative plant–AM fungal diversity relationships observed in those biomes.

If legacy effects are shaping plant–fungal richness correlations, we hypothesised that correlations would be stronger in areas with historically more stable climates, and weaker in areas with higher human land modification (Fig. 1D, E). In support of this hypothesis, we found that plant–AM fungal richness correlations were more positive in regions with more stable past climates ($p < 0.001$; Fig. 3D). However, there was no significant relationship between correlations and human development (Fig. 3E). No relationship was evident between plant–ECM fungal richness correlations and climate stability or human development.

In addition to the support for our hypothesised richness correlation drivers, plant–fungal correlations were also significantly related to some of the other environmental covariates included in the model. For example, plant–AM richness correlations were positively related to soil pH, soil organic carbon, and elevation, negatively related to soil phosphorus, and had a hump-shaped relationship with soil nitrogen (Supporting Information Fig. S5). Plant–ECM richness correlations were positively related to soil pH, soil organic carbon, and mean annual temperature, and negatively related to soil nitrogen (Supporting Information Fig. S5). More work is needed to uncover the potential mechanisms driving these patterns.

## Symbiosis effects

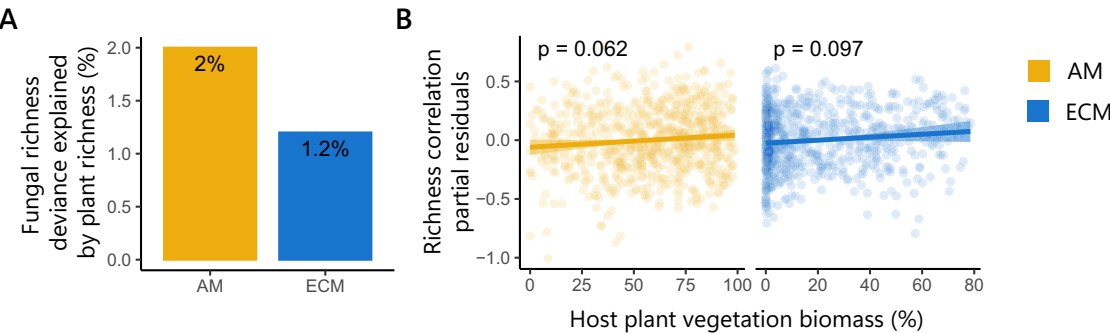

## Environmental effects

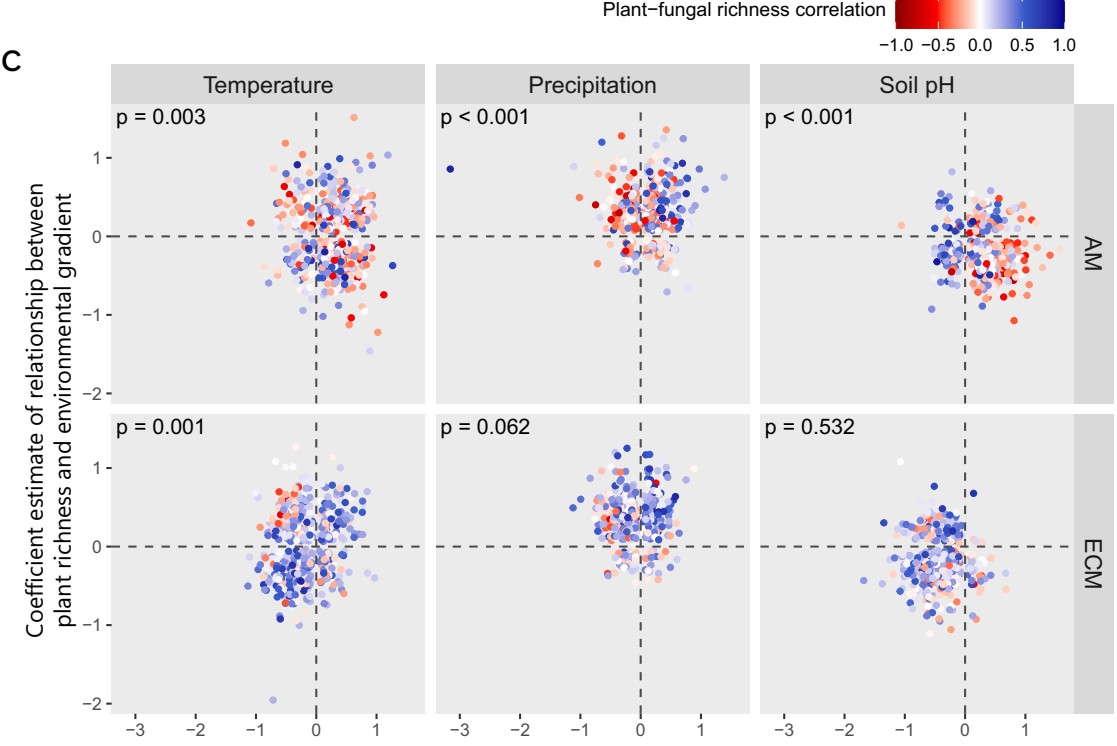

## Legacy effects

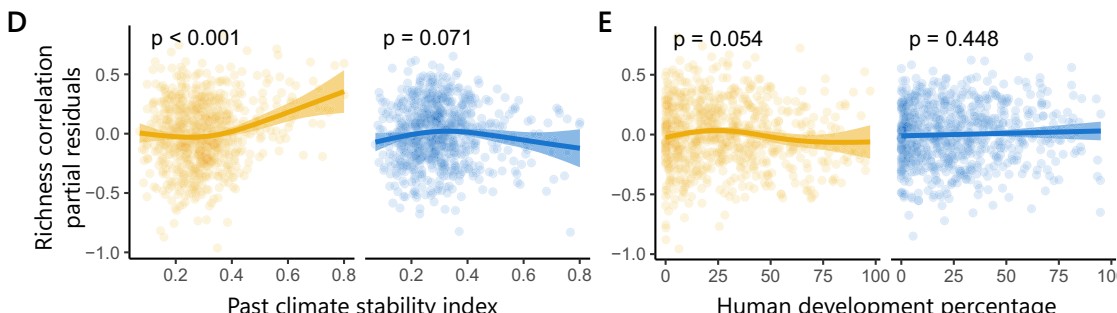

### Richness hotspot overlap

Given that the plant–fungal richness correlations were weak or negative in many regions, it is likely that there may be a mismatch in the locations of plant and fungal diversity hotspots that are potentially of high conservation value. To test this, we mapped the grid cells with richness values in the top 95th percentile (the threshold for identifying biodiversity hotspots following previous studies[26,28,30]) for each of the original richness geospatial layers. We used the individual geospatial layers rather than the consensus maps produced for the other analyses to provide a more complete picture of potential hotspot regions that incorporates variation in the richness predictions produced by different studies. Due to different underlying datasets and modelling

**Fig. 3 | Tests of correlation mechanism hypotheses described in Fig. 1.** All richness values are from the consensus richness maps produced to combine the richness predictions published by different authors (see 'Methods'). Evidence for symbiosis-driven diversity coupling effects: **A** the partial percentage deviance in fungal richness directly explained by plant richness at the global level (based on generalised additive models (GAMs) including environmental covariates using 10,000 random global grid cells, see Supporting Information Fig. S6), and **B** partial residual plots from GAMs testing how richness correlations within ecoregions (points) changes as the percentage of vegetation biomass belonging to potential host plant species[33] increases (AM host vegetation for AM fungal–plant correlations and ECM vegetation for ECM fungal–plant correlations). Evidence for environmental covariates driving diversity correlation patterns: **C** p-values show results of two-sided chi-squared tests testing the hypothesis that positive plant–fungal correlations (blue) occur in ecoregions when plant and fungal richness respond similarly (top right and bottom left panel segments) to mean annual temperature,

mean annual precipitation, and soil pH, and negative plant–fungal correlations (red) occur when they respond in opposite directions (top left and bottom right panel segments). Coefficient estimates were calculated from linear models including plant richness, fungal richness, and percentage host plant vegetation biomass as covariates to remove any variation due to potential symbiotic effects (see 'Methods'). Biome summaries are provided in Supporting Information (Fig. S4). Evidence for both long-term and short-term legacy effects influencing correlation patterns: partial residual plots for GAMs testing the relationship between richness correlations within ecoregions and **D** climate stability index[24] and **E** human development percentage (see 'Methods'). Variables shown in (**B**, **D**, **E**) were included in the same GAMs, along with other environmental covariates relating to climate, soil properties, vegetation, and species richness (see 'Methods' and Supporting Information Fig. S5). The line shows the fitted curves, and shading shows one standard error. AM arbuscular mycorrhizal fungi, ECM ectomycorrhizal fungi.

approaches, there were some differences in the exact hotspot locations identified by different studies (Supporting Information Fig. S7, Table S2). However, there was generally consensus within each taxonomic group regarding the geographical regions predicted to have the highest diversity. As shown previously[28], global plant richness hotspots are concentrated in tropical and subtropical forests of Central and South America, central Africa, Madagascar, and Southeast Asia (Fig. 4). Although AM fungal hotspots are also predominantly in tropical regions such as the Brazilian Cerrado, Guinean forests of West Africa, and Southeast Asia, the major ECM fungal hotspots are distributed mostly across temperate and boreal forests in Canada, northern and western US, central Europe, Russia, and eastern Asia[26] (Fig. 4).

By comparing hotspot overlap, we found a strong mismatch in plant–fungal hotspot locations for both mycorrhizal groups, with only 8.8% of AM fungal hotspot area and 1.5% of ECM fungal hotspot area overlapping with plant hotspots (Fig. 4). The main regions of hotspot overlap occurred in Central America and Southeast Asia for both AM and ECM fungal hotspots. However, plant–AM fungal hotspot overlap was predominantly in the tropical moist forest biome, whereas plant–ECM fungal hotspot overlap was mostly in the tropical conifer forest biome (Fig. 4). We also re-calculated hotspot overlap within each biome to determine if overlap was higher in biomes with greater host plant dominance of a particularly mycorrhizal type. Hotspot overlap was generally greater at the biome level (ranging between 1.0% and 41.3%), but there was no positive relationship between hotspot overlap and the prevalence of potential host plants (Supporting Information Fig. S8).

## Uncertainty

Modelled geospatial layers like those used in our study are associated with considerable error originating from factors such as training data sampling bias, model misspecification, and error propagation through predictive covariates[26,31]. We aimed to account for the impact of this potential error by combining diversity geospatial layers from multiple studies that used different methods and training data, and by excluding grid cells with high uncertainty in the original layers or strong disagreement between studies. We also examined how model prediction uncertainty of the geospatial layers might influence the strength and direction of correlations by recalculating richness correlations after adjusting richness values of each layer according to the prediction uncertainty layers provided by the authors (see 'Methods'). In most cases, adjusting for model uncertainty slightly reduced the strength of correlations both at the global level and within each biome. Regardless, the correlation direction (i.e. positive or negative) remained the same in all cases (Supporting Information Fig. S11).

We also used vascular plant and ECM fungal richness data from the NEON (National Science Foundation's National Ecological Observatory Network (NEON)) database of plots across the US to corroborate correlations with those calculated using ground-sourced data (see

'Methods'). Promisingly, our richness correlations and those from the NEON database were positively related (Pearson's $r = 0.37$; Supporting Information Fig. S12), although the relationship was non-significant (linear model: $t_{(1,10)} = 1.3$, $p = 0.23$, estimate $= 0.20$, 95% confidence interval $= -0.15$–$0.55$) due to the small number of ecoregions (12) that could be tested. We were not able to obtain reliable AM fungal richness data from the same plots to evaluate plant–AM fungal richness correlations.

## Discussion

By exploring the relationships between plant and fungal diversity, we show that the strength and direction of the coupling between these major taxonomic groups varies considerably in different regions across the globe. Few places exist that are hotspots of both plant and fungal richness. This spatial divergence in hotspots, combined with highly variable richness correlations, indicates that conservation strategists should generally avoid relying on plant diversity as a proxy for mycorrhizal fungal diversity. However, the high plant–fungal richness correlations at smaller spatial scales in some locations (Fig. 2) could mean that plants do provide an adequate proxy in specific situations, and protection strategies for plants aboveground might benefit mycorrhizal communities belowground via this diversity connection.

Our findings support past evidence that plant–fungal diversity relationships are often stronger at local scales[10] than at the global level[11–13]. This is consistent with the fact that both plants and fungi exhibit patterns aligned with the sharp-transition hypothesis where biodiversity patterns are delineated by ecoregion and biome boundaries[32]. Diversity associations may therefore be promoted within distinct habitat types at smaller scales but then become obscured at larger spatial scales when the differing relationships across habitats are aggregated. At the global scale, our findings contradict a recent global study by Toussaint et al.[13] which suggested that plant diversity shows no correlation with AM fungal richness, and a negative relationship with ECM fungal richness. The lack of negative relationship between plants and ECM fungi at the global level in our study is intriguing given that ECM fungi are known to dominate in less plant-diverse regions (e.g. boreal and temperate zones) and show opposite latitudinal diversity gradients to plants[12,26]. The alpha diversity geospatial layers used in this study were based on nearly 10-fold more datapoints than those used by Toussaint et al.[13], but geospatial layer prediction error could lead to a weakening of patterns using our methods. More work is needed to corroborate correlations with ground-sourced data when paired plant and fungal richness data become more readily available[14].

Although it was not possible to fully quantify the mechanisms driving plant and fungal richness correlations in our study, we found supportive evidence for all three of our hypothesised potential drivers: symbiosis effects, environmental effects, and legacy effects. AM fungi

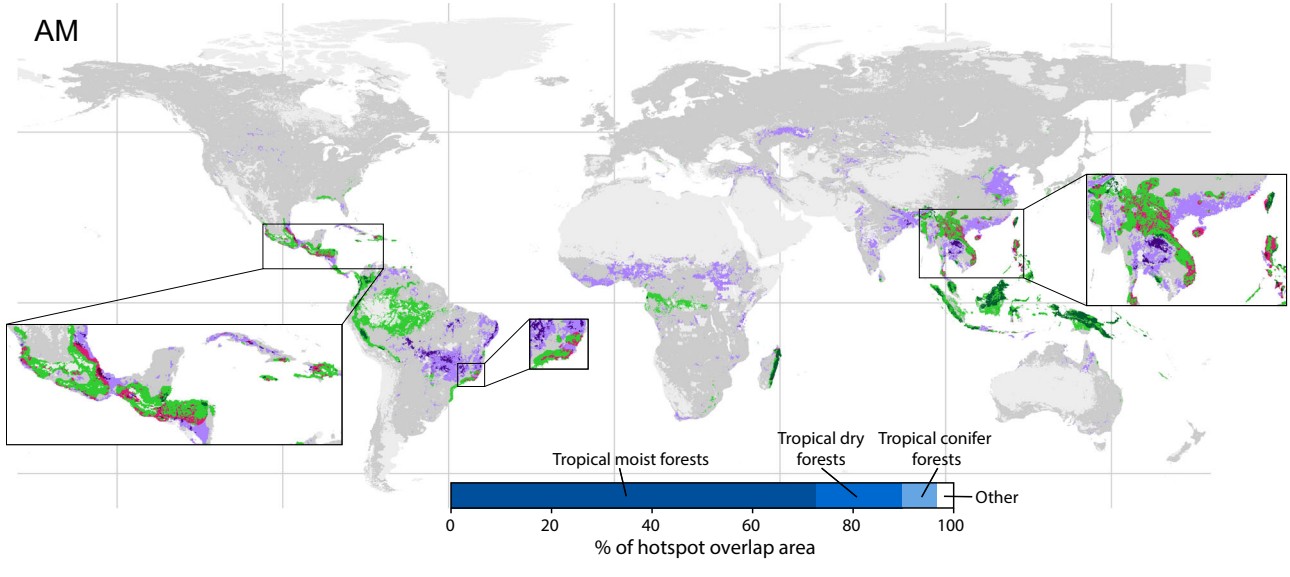

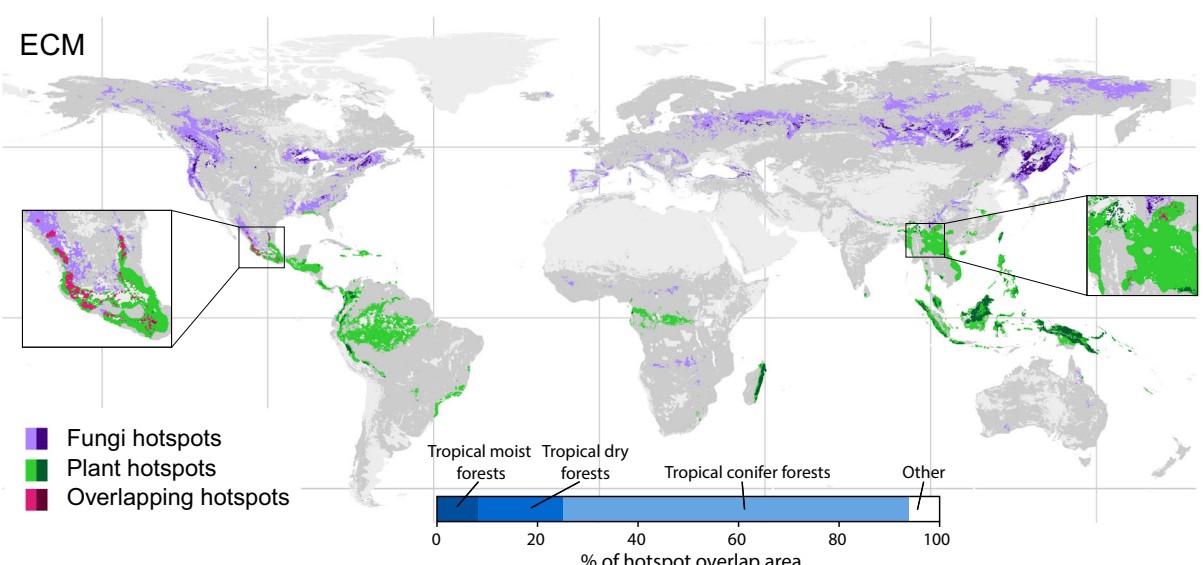

**Fig. 4 | Richness hotspot overlap.** Richness hotspots (top 95th percentile of richness predictions) for vascular plants (green), arbuscular mycorrhizal (AM) fungi and ectomycorrhizal (ECM) fungi (purple). Areas where plant and fungal hotspots overlap are shown in pink. Hotspots are mapped individually for the four previously published alpha diversity geospatial layers[26–29] (and not the consensus richness maps), and hotspot or overlap areas identified by more than one study are shaded darker. Dark grey areas are those included in the analysis; light grey areas were excluded due to high uncertainty in the original alpha diversity predictions or where predictions for each taxonomic group by different studies strongly disagreed (mostly based on the coefficient of variation between bootstrap model replicates; see 'Methods'). Blue bars show how hotspot overlap area is distributed across biomes. A breakdown of the hotspot areas for each individual geospatial layer is provided in Supporting Information Fig. S7, and maps of hotspots recalculated at the biome level (i.e. top 95th percentile within each biome) are provided in Figs. S9 and S10.

form mycorrhizal associations with a greater percentage of vascular plant species (72%) than ECM fungi (2%)[16], and so an increase in vascular plant diversity is much more likely to correspond to an increase in host plant diversity for AM fungi. Increased niche space resulting from greater host plant diversity (i.e. an increase in the host root trait space) could then lead to positive diversity coupling[14], which aligns with our result that AM fungal richness was more strongly related to plant richness compared to ECM fungi and plants. Symbiosis-driven effects could also be driving the pattern that plant–AM fungal correlations were generally positive in grassland biomes and negative in forest biomes, because this may reflect the shift in host plant dominance from AM hosts in grasslands to ECM hosts in many forests[33]. Local-scale studies support the presence of positive plant–AM fungal

diversity coupling in grasslands[34]. Conversely, ECM-dominated forests often exhibit positive plant–soil feedbacks (where ECM plants and fungi promote the establishment of other ECM species) that could exclude AM fungal species[35], producing the negative plant–AM fungal diversity relationships we observed in tropical and temperate forest biomes. However, empirical tests examining the degree to which plants and mycorrhizal symbionts are truly coupled and likely to directly influence each other's diversity are lacking[14,17]. Additionally, we found no support for our hypothesis that diversity coupling, and therefore plant–fungal richness correlations, would be stronger in ecoregions that contain vegetation dominated by host plants. Further evidence is therefore required to confirm whether the richness correlations we observed are in fact driven by symbiosis effects.

Environmental covariation was also clearly driving some diversity associations, particularly between AM fungi and plants. Different taxa can respond at different rates or in different directions to drought, heat, or other environmental stressors[36,37], and so it is unsurprising that environmental covariates could lead to both positive and negative diversity relationships arising. The significant positive relationship we observed between plant–AM fungal richness correlations and the climate stability index indicates that past evolutionary patterns may also influence diversity correlations to some degree. If fungi can evolve more quickly in response to climate fluctuations than plants[38], host–symbiont co- or correlated evolution may have been weakened in regions with unstable past climates[14,24,39], leading to weaker richness correlations in these areas. In terms of more recent legacy effects, plant diversity is often negatively affected by human-induced disturbances, but AM fungal diversity can be positively affected[25,26]. We did not observe significantly weakened or negative plant–fungal correlations when human development was higher, but it is possible that other, more specific types of disturbances could affect diversity relationships more strongly. Legacy effects are highly context-dependent and include many different disturbance types[21], and their relative importance in structuring diversity relationships likely varies regionally.

Future work is needed to untangle the mechanisms driving diversity associations in more detail. Our modelling approach allowed us to evaluate the evidence for each mechanism somewhat independently of the others, by removing variation potentially explained by alternative mechanisms (see 'Methods'). However, we acknowledge that biodiversity patterns are the outcome of multiple eco-evolutionary processes acting in concert[40]. Thus, the importance of each mechanism is likely to differ in different biogeographical regions or at different spatial scales. Future research should be undertaken to understand potential nonlinearities and context-dependencies giving rise to diversity patterns, for example using mechanistic macro-eco-evolutionary models[41,42] or controlled mesocosm experiments[17]. A mechanistic understanding of diversity relationships will become increasingly informative to conservation planning as climate change creates ecosystems and species assemblages for which few precedents exist.

The role that symbiotic relationships play in driving plant–fungal diversity correlations is particularly intriguing and has important implications for conservation and management. If diversity is strongly coupled, avoiding significant biodiversity loss in one level could have cascading benefits to the diversity and functioning of the other[43]. Whilst the fact that richness of AM fungi and plants is more tightly linked than ECM fungi and plants aligns with the idea that diversity coupling could be driving correlation patterns, more direct evidence is needed. Global geospatial layers of mycorrhizal host plant diversity are not yet available, and so we were limited to using geospatial layers of all vascular plant diversity. Therefore, we were unable to examine relationships between mycorrhizal fungal diversity and host plant diversity which would allow a more thorough investigation regarding the role of symbiotic effects in structuring diversity relationships. Interestingly, the ECM fungal hotspot areas that overlapped with vascular plant hotspots occurred mostly in central Mexican tropical conifer forests which are known to be a centre of ECM host plant phylogenetic diversity[44]. This supports the idea that diversity relationships between hosts and their mycorrhizal symbionts may be stronger than those between mycorrhizal fungi and all vascular plants. However, available evidence that mycorrhizal fungal diversity is related to host diversity is weak[12,13], so more thorough investigation is needed once more accurate host plant geospatial layers become available. Additionally, the fungal diversity geospatial layers used operational taxonomic units created from clustered sequencing data to define fungal species, which may over- or under-represent the true number of species in different fungal lineages[45]. This could lead to inaccuracies in fungal species richness calculations, thereby weakening out ability to detect plant–fungal diversity coupling patterns. Future advancements in defining fungal species will hopefully improve fungal richness data accuracy in the future.

The limited extent of the overlap between plant and fungal diversity hotspots indicates that plant hotspots are likely inadequate proxies for mycorrhizal fungal hotspots in most situations. Therefore, conservation decisions based purely on aboveground plant community assessments may be missing the most diverse areas of mycorrhizal fungi that are critical for ecosystem functioning. This aligns with more general aboveground–belowground diversity hotspot mismatches that have been previously identified[9]. Conservation and restoration research and implementation are currently heavily biased towards aboveground macro-organisms[6,46,47], which is problematic for soil organisms[9]. There have been recent calls for policymakers to better incorporate fungi and other microbes when designing conservation and management strategies[9,48,49], and our results provide added support for this. Our analyses examined hotspot overlap at the global and biome scales, and it is important to note that many conservation decisions are made at smaller local scales[50]. The stronger richness relationships identified in some ecoregions indicate that plant richness might in fact adequately represent mycorrhizal fungal richness in some cases. However, the majority of ecoregions contained weak or no richness correlations. It is also worth noting that our identification of hotspots using geospatial layers is dependent upon the accuracy of the models used to predict richness. We masked regions with high model uncertainty, and it is possible that some of these locations may also contain high species richness. Many regions are also understudied, particularly in terms of fungal diversity[51,52]. Our ability to detect fungal richness hotspots will evolve as diversity databases improve.

In conclusion, our analysis reveals that the relationship between plant and fungal diversity is highly context-dependent, and likely controlled by a range of interacting ecological mechanisms. As such, aboveground diversity may not be a good proxy for belowground diversity in guiding conservation priorities. Plant diversity hotspots show remarkably little overlap (<8.8%) with fungal diversity hotspots at the global level, suggesting that the effective protection of belowground biodiversity may require specific management consideration. Evaluating the relationships between other types of plant and fungal diversity measures, such as beta diversity, gamma diversity, phylogenetic diversity, rarity, and endemism[29,53], should be the focus of future work to provide a more complete picture of aboveground–belowground diversity linkages.

## Methods
### Alpha diversity data layers
We used previously published global predictions of AM and ECM fungal richness from Van Nuland et al.[26] and Mikryukov et al.[29], and vascular plant richness from Sabatini et al.[28] and Cai et al.[27]. Both of the fungal richness studies generated predictions at 30 arc-s resolution (~1 km$^2$ at the equator). Van Nuland et al.[26] used random forest models trained on ITS sequences for ECM fungi and SSU sequences for AM fungi from the GlobalFungi and GlobalAMFungi database[52,54]. Mikryukov et al.[29] used boosted regression trees with sequence data from the Global Soil Mycobiome consortium (https://gsmc-fungi.github.io/). For vascular plants, Sabatini et al.[28] modelled forest and non-forest vascular plant richness separately at 2.5 arc-min resolution (~4.63 km$^2$ at the equator) using boosted regression trees with data from the vegetation plot database 'sPlot' from multiple plot sizes ranging between 10 m$^2$ and 10,000 m$^2$ (www.idiv.de/splot). We selected the joined forest and non-forest prediction that was based on data from plot sizes of 1000 m$^2$. Cai et al.[27] built vascular plant richness models using the regional checklists of native vascular plants in the GIFT database[55] as training data and five modelling methods to generate consensus predictions for 7774 km$^2$ hexagons (generalised linear

models, generalised additive models (GAMs), random forest models, extreme gradient boosting, and neural networks). We selected the interpolated ensemble prediction raster (average of the five modelling methods weighted by model accuracy). Both vascular plant layers were provided by the authors resampled at 30 arc-s resolution. We reprojected all layers to Equal Earth projection at 1 km² resolution using the 'terra' package version 1.7-78[56] in R version 4.4.1[57].

Modelled geospatial layer predictions contain uncertainty from factors such as training data sampling bias, model misspecification, and error propagation through predictive covariates[26,31]. Therefore, we created uncertainty masks to exclude grid cells with either high model uncertainty in the original richness layers (as calculated by the original authors) or strong disagreement between the predictions produced by different studies. Uncertainty was measured differently across studies. Van Nuland et al.[26] quantified fungal richness uncertainty as the coefficient of variation (CV; the standard deviation divided by the mean) between 100 random bootstrap replicates (stratified across biomes), while Mikryukov et al.[29] calculated uncertainty as the standard deviation (SD) from 10-fold cross-validation. For plant diversity by Sabatini et al.[28], uncertainty was the percentage ratio between the interquartile distance (IQR) and the median of species richness estimates across 99 stratified resampled subsets of the data (using stratum based on unique combinations of realms, biomes, and plot sizes). For Cai et al.[27], uncertainty was the CV of predictions produced by the five different modelling methods. For all layers, we created masks to exclude grid cells with uncertainty values in the top 95th percentile (using the 'quantile' function from the 'terra' R package; see Supporting Information Table S3 for the 95th percentile thresholds for each layer).

We generated consensus richness predictions for the two layers in each taxonomic group by (1) log-transforming the richness values of rasters that had not previously been transformed after adding 1 as a constant (which is consistent with the approach used by Mikryukov et al.[29]), (2) standardising (i.e. centreing and scaling) each raster layer to have zero mean and unit standard deviation, and (3) calculating the mean value for each grid cell. We also calculated the CV across the two raster layers within each taxa for each grid cell (i.e. in the case of two raster layers, the CV is half the absolute difference divided by the mean). We then excluded regions with CV values in the top 95th percentile (i.e. those with predictions that strongly disagreed between the studies). Standardising the richness layers meant that the mean contained negative values which prevents calculation of the CV, and so we added a constant to both layers (the minimum value across both) to make all values positive prior to calculating the CV. Whilst the size of the constant can influence the magnitude of the CV, this is not the case when only two values are used in the CV calculation. Additionally, the rank order of CV values remains the same regardless of the constant, and this information was what was required to exclude the top 95th percentile. We chose to use the CV rather than the SD or range difference to ensure that we did not bias exclusion towards grid cells with higher mean richness values that may contain a greater SD. Grid cells excluded were those with CV values > 0.15 for vascular plants, >0.51 for AM fungi, and >0.24 for ECM fungi (Supporting Information Table S3). This means that no grid cells were included where the standard deviation of the two layers was more than 51% of the mean.

The relevant uncertainty masks were applied to all richness layers prior to extracting data and conducting the analyses described below. For plant–AM fungal comparisons, the masks included the two plant richness uncertainty masks (from Sabatini et al.[28] and Cai et al.[27]), the two AM fungal uncertainty masks (from Van Nuland et al.[26] and Mikryukov et al.[29]), regions where the two plant layers disagreed, and regions where the two AM fungal layers disagreed. Richness layers used in plant–ECM comparisons were masked in the same way but with the ECM-relevant alternatives. Therefore, plant richness layers were masked differently depending on if they were involved in AM or ECM fungal comparisons, and so statistics calculated from extracted data

differ slightly. A total of 67.8% and 68.6% of available global terrestrial grid cells remained included after applying uncertainty masks for AM and ECM fungal analyses, respectively (Supporting Information Table S4). At least 50% of grid cells were retained within most biomes (Table S4), except for Montane Grasslands (43.7% for AM analyses and 41.1% for ECM analyses) and Deserts (21.1% for AM and 32.7% for ECM) (Supporting Information Table S4). After masking, Spearman correlations between data from the two geospatial layers of each taxon ranged between 0.45 and 0.68 (Supporting Information Fig. S13).

## Richness correlations

We used Spearman rank correlation coefficients (to allow for nonlinearity in monotonic relationships) to assess alpha diversity correlations between (i) plant and AM fungal richness and (ii) plant and ECM fungal richness. For the main analysis, we calculated correlations using the consensus richness predictions created from averaging the two predictions within each taxonomic group (described above). Analysis of how correlations differed between the individual richness layers is provided in Supporting Information Fig. S14. Correlations were calculated at the global, biome and ecoregion levels[15] and based on richness values from 10,000 random grid cells selected across the globe, 10,000 cells selected within each of the 14 biomes, and up to 1000 cells selected within each of the 846 ecoregions using the 'spatSample' function from the 'terra' R package. Fewer grid cells were used for ecoregions because of their smaller size. For ecoregions with fewer than 1000 grid cells, all grid cells were used, and only ecoregions with at least 100 grid cells were included (766 ecoregions for AM analyses and 770 for ECM analyses).

We also examined plant–fungal richness relationships using GAMs, to allow for more complex non-linear relationships (e.g. quadratic relationships). We used the 'gam' function from the 'mgcv' R package version 1.9-1[58], and modelled fungal richness as a function of plant richness (separately for AM and ECM fungal richness) within each ecoregion, using the randomly selected ecoregion grid cells described above. We used a Gaussian distribution, REML as the smoothing parameter estimation method, thin plate regression splines, and three basis functions ($k$). Other parameters were set at function defaults. We compared the percentage deviance explained by the model for each ecoregion to the absolute value of the Spearman correlation coefficient described in the previous paragraph, by calculating Spearman rank correlation coefficients of the two metrics. The metrics were correlated $r = 0.86$ for the AM analysis and $r = 0.87$ for the ECM analysis (Supporting Information Fig. S3).

We examined how richness correlations might be impacted by underlying uncertainty in the original richness geospatial layers to evaluate whether error could lead to spurious correlations being identified. To do this, we used the model uncertainty layers provided by the original authors (those described above that were used to make the uncertainty masks) to evaluate how correlations change if richness values were to fall anywhere within of SD of the original model bootstrap replicates (or SD of the five model types in the case of Cai et al.[27]). When the CV was provided as the uncertainty layer, we back-transformed values to obtain the SD by dividing by 100 and multiplying by the mean. Sabatini et al.[28] provided uncertainty as the percentage ratio between the IQR and the median. Therefore, we divided by 100, multiplied by the median, and multiplied by the constant 1.35 to approximate the SD. This approximation relies on the bootstrap replicates being normally distributed which we were unable to confirm. However, this approximation was required to make this layer comparable to the others. We also treated the median richness raster from Sabatini et al.[28] as equivalent to the mean richness rasters of the other layers. We generated a distribution of possible correlations that could arise given the SD around the mean of each grid cell by (1) taking the 10,000 random grid cells of each of the four richness geospatial layers used to calculate the original correlations and adding a random

value (selected from a uniform distribution) between plus and minus the SD of the grid cell, (2) log-transforming the new error-adjusted richness values (except for the Mikryukov, et al.[29] layers that were log-transformed prior to calculating the provided mean and SD), (3) standardising the log-transformed error-adjusted values around a mean of 0 and SD of 1 (based on the range of values present in the full raster layer, not just the 10,000 random cells), (4) calculating the average of the adjusted values of the two geospatial layers for each taxonomic group, and (5) calculating Spearman rank correlation coefficients between the adjusted plant and fungal richness values. We repeated these steps for both plant–AM and plant–ECM fungal richness correlations 1000 times and examined the distribution of correlation values returned. We did this both at the global level and within each of the biomes. As expected, results showed that adding error generally weakened correlations, but no iterations produced correlations with the opposite sign to that calculated in the original analysis (Supporting Information Fig. S11).

We also evaluated how the correlations calculated within ecoregions compare to those calculated using ground-sourced data. Paired plant and fungal richness data are rare at the global level, but continental-scale data on vascular plant and ECM fungal richness is available from the US NEON. Unfortunately, we were unable to obtain reliable AM fungal richness data from the same plots. We obtained the processed ECM fungal richness data from Qin et al.[59], and matched that with vascular plant data for the same plots downloaded from the 'plant presence and percent cover' data product (DP1.10058.001) using the 'neonUtilities' R package version 2.4.2[60]. We counted the number of unique taxon IDs present within each plot as the plant species richness. Data were available for 426 plots from 28 ecoregions across the US. However, most ecoregions contained limited replication, and so we calculated Spearman plant–fungal correlations for the 12 ecoregions that contained at least 10 plots. NEON plot plant–fungal richness correlations were positively related to those calculated in this study (Pearson's $r = 0.37$, Supporting Information Fig. S12), providing support that our geospatial layers are approximating on-the-ground patterns. However, this relationship was non-significant (linear model: $t_{(1,10)} = 1.3$, $p = 0.23$, estimate = 0.20, 95% confidence interval = −0.15–0.55) due to the small number of ecoregions tested.

## Mechanisms driving richness correlations

We performed multiple analyses to look for evidence supporting the potential mechanisms driving correlation patterns outlined in Fig. 1. As a measure of host plant prevalence (Fig. 1B), we used geospatial layers of the percentage of aboveground plant biomass belonging to AM or ECM host plants[33]. For analyses relating to past climate stability, we used the climate stability index[24] which maps the variability of bioclimatic variables between the Pliocene (3.3 Ma) and the present. To assess human disturbance effects, we followed Van Nuland et al.[26] and summed the geospatial layers from EarthEnv[61] of the percentage of each grid cell covered by urban/built-up areas and cultivated/managed vegetation to create a measure of 'human development percentage'. This metric provides an indication how disturbed the landscape is by both urbanisation and human agricultural/land management activities. Other environmental geospatial layers included mean annual temperature, mean annual precipitation (both from CHELSA[62,63]), soil pH, soil organic carbon content (SOC), soil nitrogen (all from the upper 5 cm of soil[64]), Olsen plant-available phosphorous[65], and elevation (EarthEnv[66]). All layers were reprojected to Equal Earth projection at 1 km² resolution to match the alpha diversity layers.

**Symbiosis effects and legacy effects.** To test whether plant–fungal symbiosis could be driving correlations, we calculated the percentage of variation in fungal richness that could be directly attributed to changes in plant richness after accounting for variation explained by

environmental covariates (as a test of Fig. 1A). We conducted GAMs to account for non-linearity between fungal richness and some environmental variables (using the 'mgcv' R package version 1.9-1[58]; Fig. S6). We used values from the 10,000 randomly selected global grid cells, and modelled fungal richness (separately for AM and ECM) as a function of plant richness including temperature, precipitation, pH, SOC, soil nitrogen, soil phosphorus and elevation as covariates. These covariates were chosen due to their expected influence on plant and fungal richness patterns and their availability as global geospatial layers. We used a Gaussian distribution, REML as the smoothing parameter estimation method, thin plate regression splines, and three basis functions ($k$). Other parameters were set at function defaults. We used the 'gam.check' function to check model diagnostic plots and assist with selecting $k$. Percentage deviance in fungal richness explained by plant richness within each model was calculated as follows based on Brunbjerg et al.[67]:

$$\text{Deviance explained} = \frac{\text{deviance}(\text{reduced model without plant richness}) - \text{deviance}(\text{full model})}{\text{deviance}(\text{null model with intercept only})}$$

(1)

Additionally, we used GAMs to model how correlations within each ecoregion vary according to the prevalence of potential host plants (as a test of Fig. 1B), or legacy effects relating to past climate stability and human disturbance (tests of Fig. 1D, E). Plant–AM fungal and plant–ECM fungal richness correlations at the ecoregion level were used as response variables, and were modelled as a function of aboveground host plant vegetation biomass (AM plant biomass for the AM model and ECM plant biomass for the ECM model), the climate stability index, and the human development percentage. Covariates of temperature, precipitation, pH, SOC, soil nitrogen, soil phosphorus, elevation, raw fungal richness and raw plant richness were also included to account for variation explained by non-symbiosis or non-legacy effects. Covariate variable data were ecoregion means of the same randomly sampled grid cells within each ecoregion (up to 1000) that were used to calculate richness correlations. Correlations between covariates were <|0.64|, except for precipitation and soil pH ($r = −0.73$), precipitation and plant diversity ($r = 0.75$), SOC and soil nitrogen ($r = 0.76$), and ECM fungal richness and ECM host vegetation percentage ($r = 0.79$). GAM parameters were as described in the previous paragraph, except that four basis functions were used.

**Environmental effects.** We tested whether the direction of plant–fungal richness correlations (positive or negative) could be influenced by underlying relationships between richness and environmental covariates by examining whether positive plant–fungal correlations are more likely to occur when plant and fungal richness respond similarly to environmental gradients, and negative correlations are more likely when richness responses differ. Within each ecoregion we fitted two linear models – one modelling plant richness and one modelling fungal richness—to extract model estimates of the relationship between richness and environmental gradients. We chose linear models instead of GAMs so that we could extract directional (positive or negative) relationship estimates. We tested mean annual temperature, mean annual precipitation, and soil pH. We did not test SOC, soil nitrogen, soil phosphorus, or elevation due to high collinearity between these and other variables within many ecoregions. Richness of the opposite group (fungal richness for plant models and plant richness for fungal models) and the percentage of plant biomass belonging to potential host vegetation[33] were included as covariates to remove variation due to potential host–symbiont coupling effects. Environmental data came from the same randomly selected grid cells (up to 1000) within each ecoregion that were used to calculate plant-fungal richness correlations. In addition to excluding ecoregions with fewer than 100 grid cells, we

excluded ecoregions where environmental predictors contained fewer than 10 unique values (700 out of 867 ecoregions remained). We confirmed that response variables met normality assumptions in all ecoregions by flagging potentially skewed richness data using the 'shapiro.test' R function from the 'stats' package version 4.4.1[57] and checking histograms. All variables were standardised to range between 0 and 1 within each ecoregion prior to conducting the model so that model estimates were comparable between variables.

We extracted the model coefficient estimates for each ecoregion, and then performed Chi-squared tests ('stats' R package version 4.4.1) for each environmental variable to test the hypothesis that ecoregions where plant and fungal richness were related to that variable in the same direction (i.e. coefficient estimates were either both positive or both negative) contained positive plant–fungal correlations, and ecoregions with opposite model estimates contained negative plant–fungal correlations. Only coefficient estimates with $p$ values < 0.05 were included in Chi-squared tests because non-significant estimates were considered unreliable. Some ecoregions had high covariate collinearity; 9.0% (AM models) and 9.1% (ECM models) of ecoregions contained covariates with Pearson's $r$ correlations > |0.85|, and 20.3% (both AM and ECM) contained correlated covariates of $r >$ | 0.8|. We recalculated Chi-squared values after removing these ecoregions, and patterns remained relatively stable (Supporting Information Figs. S15 and S16), indicating covariate collinearity did not strongly influence the overall pattern.

### Richness hotspots

Alpha diversity hotspots were defined as the grid cells with richness values greater than the upper 95th percentile, and were calculated separately for the two geospatial layers within each taxonomic group after applying the relevant uncertainty masks (described above). 95th percentile cut-off values were calculated using the 'quantile' function from the 'terra' R package. We also calculated hotspots using 80th, 85th, 90th, and 98th percentile cut-off values to see how hotspots varied when a different threshold was chosen (Supporting Information Fig. S17). We chose to use the 95th percentile threshold in the main analysis because this follows methods used by past studies[26,28,30]. There were some differences in the exact grid cells identified as hotspots between the two layers (Fig. 1 and Supporting Information Fig. S7 and Table S2), however they generally reflected the same geographical regions. All grid cells identified by either one or both studies were considered hotspots and were used to assess hotspot overlap between (i) plants and AM fungi and (ii) plants and ECM fungi. We mapped hotspots and calculated hotspot overlap both at the global level and within each of the 14 biomes.

### Reporting summary

Further information on research design is available in the Nature Portfolio Reporting Summary linked to this article.

## Data availability

The plant and fungal richness geospatial layers used in this study are available from Sabatini et al.[28], Cai et al.[27], Van Nuland et al.[26], and Mikryukov et al.[29]. Data extracted from these layers used to produce the figures is available on Figshare at https://doi.org/10.6084/m9.figshare.28082948.

## Code availability

Code used to conduct the analyses and produce the figures is available on Figshare at https://doi.org/10.6084/m9.figshare.28082948.

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

## Acknowledgements

We thank Lirong Cai and Francesco Sabatini for providing access to additional plant diversity uncertainty geospatial layers, and Johan van den Hoogen and Gabriel Smith for statistical advice. The Society for the Protection of Underground Networks (SPUN; authors L.G.v.G., J.D.S., C.Q., A.C., B.F.M., E.T.K. and M.V.N.) is supported by the Jeremy and Hannelore Grantham Environmental Trust, the Paul Allen Family Foundation (202406-15969), and the Schmidt Family Foundation (G-22-63517). In addition, J.D.S. was supported by NWO Gravity Grant MICROP (024.004.014), J.D.S. and E.T.K. were supported by an NWO-VICI grant (202.012), the Human Frontier Science Program (HFSP; RGP 0029) and the Ammodo Foundation, and L.G.v.G. and T.W.C. were supported by the Bernina Foundation (2022-FS-318) and DOB Ecology (EC-2021-GEM003).

## Author contributions

L.G.v.G., M.E.V.N., T.W.C. and E.T.K. designed the project. L.G.v.G. conducted the analyses with advice from M.E.V.N., T.W.C., J.D.S., and C.Q. L.G.v.G. led the writing of the manuscript with input from all other authors including A.C. and B.F.M.

## Funding

## Competing interests

The authors declare no competing interests.
