## [Peer Review file · Nature Communications]

Global divergence in plant and mycorrhizal fungal diversity hotspots

Corresponding Author: Dr Laura van Galen

Version 0:

Reviewer comments:

Reviewer #1

(Remarks to the Author)

This study focused on the topic of plant-fungal diversity relationship--which is always concerned and has great significance in ecology, and presented a global map on the diversity hotspots and their richness correlations, as well as potential drivers behind such correlations. The authors did a good job in data analyses, figure performance and writings. This work is important for understanding the global patterns of plant-microbes interactions, and I like the mechanisms the authors introduced here: symbiosis effect, environmental effect and legacy effects. Overall, this work has great value for understanding aboveground and belowground diversity relationships, which is worth published. However, some concerns need to be addressed before that.

General comments :

1. Although the diversity hotspots defined here (the grid cells with richness values greater than the upper 95th percentile) is generally OK, Is that mean the real hotspot? I note that some areas such as tropical Tanzania, Tibetan Plateau in China were not included. The inferred conclusion only based on published data.
2. Please added some discussions related to the limitation of this study. In my opinion, if there was great divergence between plant and mycorrhizal fungal diversity hotspots, why they are significantly and positively correlated with each other, although scale is regional. In other words, the global pattern of correlation might be not very important, because the conservation and prediction is always regional and local from the government policy, and as such it is targeted. While the hotspots identified is necessary I think.

Specific comments:

1. Which is human disturbance factors ? please add more details in the Methods.
2. Can you explain more background of the term "ecoregion". Readers may have no concept for this word.
3. Please add the significance P values in the Figure 2 and Fig. S9, and checked the whole manuscript.
4. The descriptions of Table S2 legend are not very clear.
5. I did not see other important soil properties related C, N, P which may affect plant-AMF richness correlations in grasslands.
6. Please added more details in GAM and hotspots mapping analyses, such as R packages. what is the basis function and link function in GAM.

Reviewer #2

(Remarks to the Author)

This manuscript investigates links among aboveground plant and belowground mycorrhizal fungal alpha diversity patterns at the global, biome, and ecoregion scales. Overall, the ideas in this manuscript and the quest to find universal hotspots of diversity are admirable. Moreover, this analysis is one of the most data-rich and detailed comparisons of plants and fungi at the global scale. However, this paper contains many analyses, some on the same dataset and some on subsets of data conducted in different ways. It is a huge feat to explain these nuances to the reader which leads to many inconsistencies that make it difficult to understand patterns in diversity for both plant and mycorrhizal fungal groups. I have several detailed comments on this below.

(1) It is unclear when multiple plant and AM fungal layers from different data sources are combined and when they are not. Are all figures in the main text on consensus maps of plants and fungi? If so, clearly state this in the figure legends. The discrepancy amongst datasets in the supplement is alarming given the global nature of all the maps. I'm curious how much of the discrepancy among datasets is due to differential sampling coverage and depth versus differences in species definitions. Are datasets within guild rarefied to the same sampling depth? If not, why combine the two?

(2) Species definitions

I expected some discussion of the differences in species definitions among plants, AM fungi, and EM fungi and how that may affect "coupling" of plant and fungal diversity. For example, the Van Nuland reference for AM fungi groups fungi at $q=0$, but fungal definitions at $q=0$ may be more equivalent to plant genotypes. Similarly, many debates still occur about species definitions of AM v. EM fungi given the gene regions sequenced. Is this why you chose Spearman rank correlations instead of Pearson correlations which would allow you to see the magnitude of both datasets? Why is the Pearson correlation used for the ground-truthed data from NEON compared to your maps instead of Spearman in Figure S14? Some discussion of how all these factors may affect above-belowground coupling are warranted.

(3) Statistical inconsistencies

It is also unclear when/how shared plant-mycorrhizal fungal environmental drivers affect the correlation between plant and fungal diversity. I assume Figure 2 does not account for these shared environmental drivers, but Figure 3 does. You've laid out a nice framework for different forms of coupling in Figure 1 but the results in Figure 2 don't clearly connect to that framework, so it's unclear what you're trying to show and why it's different than Figure 3. The partial residual graphs in Figure S2 make this more confusing because they don't show how you accounted for covariation among environmental drivers and among environmental drivers and plant diversity. A SEM approach or something similarly hierarchical would be more robust and interpretable.

Are these shared environmental drivers why the direct effect of plants on fungal diversity is so low in figure 3A? Shouldn't that effect be even lower if you were showing the adjusted R²? Please include adjusted R² and degrees of freedom for every P value in Figures 2 and 3.

Additionally, why are GAMs fit in Figure 3 data to capture non-linearity whereas analysis in Figure 2 only accounts for linear fits?

Finally, how do these shared environmental effects on plants and fungi partition out at different spatial scales and is autocorrelation in environmental drivers why patterns vary across scales in Figure 2?

(4) It is unclear how much patterns of above-belowground coupling are influenced by the inclusion of ecosystems with very few EM host plants. Wouldn't a more interesting and realistic comparison be AM fungal diversity and AM plant diversity versus EM fungal diversity and EM plant diversity? Why include plant diversity estimates for plants that are not colonizable by a given guild of fungi? Accounting for this by estimating percent cover of AM versus EM plants helps, but doesn't really resolve the issue given that some ecosystems can be dominated by one EM plant species and whereas others could have the same percent cover of EM plants but be much more diverse.

Minor comments

Figure 1B. The text in lines 35-37 conflates plant diversity and plant prevalence. Then the graph and the text in line 38 switches from host diversity to host prevalence, which sounds like cover/biomass instead of diversity. I think the coupled diversity argument is much stronger given the niche space expansion angle you cite. Regardless, please modify the sentence in lines 35-37 to separate these ideas.

Figure 2. It's confusing to have biomes labeled as numbers in figure 2B and 2E. Label with the biomes instead. Similarly, Figure S3 would be more interpretable with biome designations in the figure.

Figure 3C. Make sure the values on the axes have the same significant digits. It would also be nice to have these panels on the same scale on the x and y axes to facilitate comparisons. Similar supplemental figures should follow suit.

Figure 4. This figure is not resolved enough to see the grid cells that are shaded darker to indicated consensus among datasets.

Figure S2. It would be helpful here to report the adjusted R² as well as the deviance explained.

Ln 216. Change form to from.

Version 1:

Reviewer comments:

Reviewer #1

(Remarks to the Author)

Good job. No comments now

Reviewer #2

(Remarks to the Author)

The authors addressed all of my previous comments. This will be an important contribution to the literature on plant-mycorrhizal fungal distributions.

We are grateful for the suggestions provided by the Reviewers to improve the clarity of the manuscript and strengthen the results. We have revised the manuscript to address the comments which we outline in detail below. Please note that line numbers refer to the track-changes document when "all-markup" is visible.

Reviewer #1 (Remarks to the Author):

This study focused on the topic of plant-fungal diversity relationship--which is always concerned and has great significance in ecology, and presented a global map on the diversity hotspots and their richness correlations, as well as potential drivers behind such correlations. The authors did a good job in data analyses, figure performance and writings. This work is important for understanding the global patterns of plant-microbes interactions, and I like the mechanisms the authors introduced here: symbiosis effect, environmental effect and legacy effects. Overall, this work has great value for understanding aboveground and belowground diversity relationships, which is worth published. However, some concerns need to be addressed before that.

Response: Thank you for your comments. We address your concerns in detail below.

General comments :

1. Although the diversity hotspots defined here (the grid cells with richness values greater than the upper 95th percentile) is generally OK, Is that mean the real hotspot? I note that some areas such as tropical Tanzania, Tibetan Plateau in China were not included. The inferred conclusion only based on published data.

Response: We agree. To further our hotspot analyses, we have now created maps of hotspots for AM fungi, ECM fungi and vascular plants using a wider variety of percentile thresholds (ranging between the 80th and 98th percentiles) showing how the regions defined as hotspots would change under alternative cut-offs (Supporting Information Figure S17). This helps to provide greater detail of potential hotspot locations.

In addition, we have broadened the Discussion (lines 358-362) to acknowledge that the hotspots identified in our study are dependent on the accuracy of the geospatial richness models. We also acknowledge that the areas we masked out due to high uncertainty may also be hotspots, and that geographical research gaps in many locations could mean that our ability to detect hotspots is diminished in these areas. We note in the last paragraph of the Discussion that we only consider species richness hotspots in this analysis, whereas other types of biodiversity hotspot research sometimes incorporate species rarity and/or environmental threats, which may identify other locations not highlighted here.

2. Please added some discussions related to the limitation of this study. In my opinion, if there was great divergence between plant and mycorrhizal fungal diversity hotspots, why they are significantly and positively correlated with each other, although scale is regional. In other words, the global pattern of correlation might be not very important, because the conservation and prediction is always regional and local from the government policy, and as such it is targeted. While the hotspots identified is necessary I think.

Response: We have now better addressed the limitations of our study by adding much more nuanced discussion of the ability of plant richness to act as a proxy for mycorrhizal fungal richness, including discussion of how patterns change across scales (lines 353-362). We state that conservation decisions are usually made at small local scales, and that the positive correlations identified in some ecoregions could mean that plant richness is a good proxy for mycorrhizal fungal richness in specific locations (lines 256-259 and lines 353-358). However, many ecoregions contained no or weak richness relationships, and so using plant diversity as a proxy for mycorrhizal fungal diversity remains problematic in these places.

We have also added more discussion of limitations in response to Reviewer 2 regarding delimiting fungal species based on OTUs. We discuss how this could over- or under-represent diversity in some fungal groups, and acknowledge that this could lead to inaccuracies in the fungal richness data that weaken diversity correlations detected (lines 338-343).

Specific comments:

1. Which is human disturbance factors? please add more details in the Methods.

Response: We have now expanded the description of the "human development percentage" variable in the Methods (lines 518-520). It is a combination of the degree of urbanization and agricultural/land management activities.

2. Can you explain more background of the term "ecoregion". Readers may have no concept for this word.

Response: We have added a more detailed explanation of what an ecoregion is, including referencing the paper where ecoregions are defined, in the first paragraph of the Introduction.

3. Please add the significance P values in the Figure 2 and Fig. S9, and checked the whole manuscript.

Response: We chose to only present Spearman correlation coefficients in Figure 2 and Figure S9 (which is now Figure S13) because p values are highly dependent on the number of data points included. Given that we extracted values from 10,000 random grid cells, all p values were highly significant and not ecologically informative. We have now stated that in the Figure captions of Figure S13, as well as Figures S5 and S6.

However, we did include a new p value for Figure S12 (comparison between correlations from our analysis and those from ground-sourced NEON data). We included this in the Figure caption, Methods, and Results.

4. The descriptions of Table S2 legend are not very clear.

Response: We agree that this description was unclear. We have now edited it to explain that hotspot area was defined separately for each geospatial richness layer (i.e., the two plant layers, two AM fungi layers, and two ECM fungi layers). We have also edited the description of a) and b) so that it is clearer exactly what the statistics represent in each part.

5. I did not see other important soil properties related C, N, P which may affect plant-AMF richness correlations in grasslands.

Response: We did include soil organic carbon. As suggested, we have now included soil nitrogen and soil phosphorous as covariates in relevant models as well. This changed the results slightly of one of our tests of the role of symbiotic effects in structuring correlation patterns (Figure 3B). Previously, there was a positive relationship between plant-ECM fungal richness correlations and the percentage of vegetation biomass belonging to ECM host plants, matching our hypothesis that diversity coupling (and therefore plant-fungal richness correlations) would be stronger in ecoregions where host plants are more dominant. This relationship became non-significant when including soil N and P as covariates. We have updated Figure 3 and the Results accordingly, and added a sentence to the Discussion so that it is clear we did not find evidence for this hypothesis (lines 292-294).

The climate and soil environmental covariates that we included in models did not form part of our main hypotheses, but the relationships between these and plant-fungal richness correlations may still be of interest to readers. Therefore, we have added a short paragraph to the Results outlining the main patterns (lines 154-161).

6. Please added more details in GAM and hotspots mapping analyses, such as R packages. what is the basis function and link function in GAM.

Response: We have now ensured that reference to R packages and their versions are included throughout the whole Methods, including the R package used for the GAMs. We have added that models were performed with a Gaussian distribution, and that all non-specified parameters were set at function defaults. This section also includes information on the basis functions and smoothing parameter estimation method (lines 535-538).

Reviewer #2 (Remarks to the Author):

This manuscript investigates links among aboveground plant and belowground mycorrhizal fungal alpha diversity patterns at the global, biome, and ecoregion scales. Overall, the ideas in this manuscript and the quest to find universal hotspots of diversity are admirable. Moreover, this analysis is one of the most data-rich and detailed comparisons of plants and fungi at the global scale. However, this paper contains many analyses, some on the same dataset and some on subsets of data conducted in different ways. It is a huge feat to explain these nuances to the reader which leads to many inconsistencies that make it difficult to understand patterns in diversity for both plant and mycorrhizal fungal groups. I have several detailed comments on this below.

Response: We greatly appreciate your comments and have incorporated your suggestions to help make the analyses clearer and more consistent. See our detailed responses below.

(1) It is unclear when multiple plant and AM fungal layers from different data sources are combined and when they are not. Are all figures in the main text on consensus maps of plants and fungi? If so, clearly state this in the figure legends. The discrepancy amongst datasets in the supplement is alarming given the global nature of all the maps. I'm curious how much of the discrepancy among datasets is due to differential sampling coverage and depth versus differences in species definitions. Are datasets within guild rarefied to the same sampling depth? If not, why combine the two?

Response: Thank you for highlighting this area to clarify. All figures and analyses in the main text are done using the consensus maps, except for the analysis of hotspot overlap. We have now specified in the text of the "Richness correlations across scales" Results section (where the geospatial layers are first mentioned) that alpha diversity layers were combined to create consensus richness maps. Additionally, we have explained in the captions of all figures if the richness values are from the consensus maps or individual geospatial layers. We have also added a sentence to the "Richness hotspot overlap" Results section to explain that we used the individual layers and not the consensus map so that we could provide a more complete picture of potential hotspot regions that incorporates variation in the underlying richness models.

Both Van Nuland et al (2024) and Mikryukov et al (2023) (who produced the fungal diversity geospatial layers) accounted for differences in sample sequencing depth when producing predictions. All maps were normalized before creating consensus maps, so any differences in rarefaction methods do not affect combining the predictions.

Unfortunately, it is not possible for us to evaluate the extent to which different methods used by the different studies could be influencing the discrepancies in the exact grid cells that fall within the top 95th percentile of richness predictions and were therefore classified as hotspots. Given the different modelling methods and underlying datasets used by different authors to create the richness predictions, it is unsurprising that there are discrepancies. However, the two fungal richness maps available were positively correlated for both AM and ECM fungi ($r > 0.63$; line 441 and Figure S13) and the hotspot locations were

within the same geographical area (lines 196-197) which shows a general level of alignment between predictions. Future modelling work to produce more accurate richness predictions will hopefully help to resolve current discrepancies.

(2) Species definitions

I expected some discussion of the differences in species definitions among plants, AM fungi, and EM fungi and how that may affect “coupling” of plant and fungal diversity. For example, the Van Nuland reference for AM fungi groups fungi at $q=0$, but fungal definitions at $q=0$ may be more equivalent to plant genotypes. Similarly, many debates still occur about species definitions of AM v. EM fungi given the gene regions sequenced. Is this why you chose Spearman rank correlations instead of Pearson correlations which would allow you to see the magnitude of both datasets? Why is the Pearson correlation used for the ground-truthed data from NEON compared to your maps instead of Spearman in Figure S14? Some discussion of how all these factors may affect above-belowground coupling are warranted.

Response: We agree that AM and ECM fungal OTUs used by the studies producing the richness predictions may not represent true fungal species, and that the gene regions sequences may lead to over-splitting or under-splitting some fungal groups. However, because we used previously-published richness map layers we were dependent on the definitions used by the authors (which follow commonly accepted standards of fungal species definitions from clustered sequencing datasets). We have now added sentences to the Discussion to explain that inaccuracies in fungal species definitions could weaken our ability to detect plant-fungal diversity relationships (lines 338-343).

Prior to creating the consensus maps, we log-transformed map layer values so that they were normally distributed, and then standardised values to have zero mean and unit standard deviation (see lines 408-412). This ensured that any differences in the magnitude of richness values between same-taxa studies, or differences in the magnitude of values between plant and fungal predictions due to differences in species definitions, did not impact results. We also used Spearman rank correlations instead of Pearson to allow for monotonic non-linear relationships between plant and fungal richness, which we now state in line 444.

We did in fact use Spearman rank correlations to calculate richness correlations from the NEON data, and have now made this clear in the figure caption and axis labels of Figure S12.

(3) Statistical inconsistencies

It is also unclear when/how shared plant-mycorrhizal fungal environmental drivers affect the correlation between plant and fungal diversity. I assume Figure 2 does not account for these shared environmental drivers, but Figure 3 does. You’ve laid out a nice framework for different forms of coupling in Figure 1 but the results in Figure 2 don’t clearly connect to that framework, so it’s unclear what you’re trying to show and why it’s different than Figure 3. The partial residual graphs in Figure S2 make this more confusing because they don’t show how you accounted for covariation among environmental drivers and among environmental drivers and plant diversity. A SEM approach or something similarly hierarchical would be more robust and interpretable.

Response: Yes, Figure 2 does not account for drivers of correlations – it simply presents the raw correlations observed. This answers Aim 1 of our paper, which was to show how plant-fungal correlations vary at global, biome and ecoregion scales. We feel that it is important to first show how correlations vary geographically and across scales before analysing the potential drivers of correlation patterns. We have now added a new opening sentence to the “Potential drivers of richness correlations” which better links the first section about raw correlations to the second section about testing correlation drivers after accounting for shared environmental effects (lines 121-122).

We have investigated SEM and other approaches for the analysis calculating the direct influence of plant richness on fungal richness (Figure 3A and formally Figure S2 – now Figure S6). However, we believe that

our original approach of using a GAM and extracting the partial deviance in fungal richness explained by plant richness is robust and consistent with the other analyses conducted in the manuscript. In particular, our approach does explicitly address the issue raised here: by including environmental covariates in the model, and then using the equation in line 541 to extract the partial deviance explained by plant richness, this allowed us to calculate the deviance explained by plant richness that is independent of deviance related to any of the environmental covariates. This naturally accounts for covariation among environmental drivers and among environmental drivers and plant diversity, and therefore we do not believe an alternative hierarchical modeling approach like SEM is necessary. Since this may have been unclear in the original submission, we have now clarified that the GAMs included environmental covariates in the caption of Figure 3.

Are these shared environmental drivers why the direct effect of plants on fungal diversity is slow low in figure 3A? Shouldn't that effect be even lower if you were showing the adjusted R2? Please include adjusted R2 and degrees of freedom for every P value in Figures 2 and 3.

Response: In the case of Gaussian GAMs (which we have now made it clear in the methods that we are using these), deviance explained and adjusted R2 are equivalent. However, we have now included adjusted R2 and degrees of freedom for all GAM models run so that there is no confusion (see figure captions of Figures S5 and S6).

Additionally, why are GAMs fit in Figure 3 data to capture non-linearity whereas analysis in Figure 2 only accounts for linear fits?

Response: As described in our response above, we used Spearman rank correlation coefficients in Figure 2 to answer Aim 1 of our paper (how plant-fungal correlations vary at global, biome and ecoregion scales). We chose to use rank correlations here, because a) they allow us to examine whether relationships are consistently positive or negative, and b) examining correlation coefficients is a common method used by many studies to assess diversity relationships. By using rank correlations, we were able to account for any monotonic non-linear diversity relationships. We chose to use GAMs when testing potential mechanisms driving correlation patterns because it was important to account for potential non-monotonic non-linear relationships between correlations and environmental drivers.

To bridge the gap in methods between Figures 2 and 3, we have now replicated the ecoregion-scale maps in Figure 2 (Supporting Information Figure S2) using deviance explained from a GAM rather than Spearman correlation coefficients. To do this, we ran GAMs for each ecoregion with fungal richness as the response variable, and plant richness as the predictor variable. We extracted the model deviance explained, and plotted ecoregion polygons coloured according to percent deviance explained. We compared the percent deviance explained values to the absolute values of the original Spearman correlation coefficients, and they were correlated $r = 0.86$ for the AM analysis and $r = 0.87$ for the ECM analysis. We have kept the Spearman correlation coefficients as the main Figure 2 so that it is possible to examine the direction (positive versus negative) of the relationships, but we have put the new figures in the Supporting Information (Figures S2 and S3).

Finally, how do these shared environmental effects on plants and fungi partition out at different spatial scales and is autocorrelation in environmental drivers why patterns vary across scales in Figure 2?

Response: Thank you for raising this interesting point. We performed additional analyses to determine how the influence of environmental variables on plant-fungal correlations changes across scales. We replicated the ecoregion-scale GAM modelling plant-fungal richness correlations as a function of environmental covariates (the analysis shown in Supplementary Figure S5 and Figure 3B, D and E) at the biome scale. The original analysis uses ~800 ecoregions as data points, with the response variable being the plant-fungal

richness correlation, and predictor variables being the mean of environmental covariate values from random grid cells in each ecoregion. We attempted to replicate this at the biome scale (using 14 data points, one for each biome), and then calculate the partial deviance explained by each predictor variable. The aim was to see how the relative importance of each predictor variable changed between the biome-scale and ecoregion-scale analyses, i.e., to see if some variables were more important at the biome scale, and other variables more important at the ecoregion scale. Unfortunately, 14 datapoints was not enough to confidently fit a GAM. Even when reducing the number of predictor variables to three or four, diagnostic plots showed bad model fits and that overfitting was an issue. Therefore, we are unable to include this analysis or interpret these patterns to quantitatively address this point.

We agree that understanding how correlation drivers change across scales is important, but we cannot think of a robust way to test this question using currently available data. We are willing to try an alternative analysis if the reviewer has specific ideas, but for now we have qualitatively addressed this point by specifying this limitation in the Discussion. As discussed in the fifth paragraph of the Discussion (lines 311-321), understanding the relative importance of the different potential mechanisms driving correlation patterns requires substantial additional future work. We have updated this paragraph to mention that the relative importance of mechanisms will likely change at different spatial scales as well as across different geographical regions, and future work is needed to test this.

(4) It is unclear how much patterns of above-belowground coupling are influenced by the inclusion of ecosystems with very few EM host plants. Wouldn't a more interesting and realistic comparison be AM fungal diversity and AM plant diversity versus EM fungal diversity and EM plant diversity? Why include plant diversity estimates for plants that are not colonizable by a given guild of fungi? Accounting for this by estimating percent cover of AM versus EM plants helps, but doesn't really resolve the issue given that some ecosystems can be dominated by one EM plant species and whereas others could have the same percent cover of EM plants but be much more diverse.

Response: We agree that examining relationships between fungal diversity and diversity of host plants only would be a much stronger test of diversity "coupling". However, no global maps of mycorrhizal host plant diversity are currently available. Therefore, we chose to focus this paper on understanding diversity "correlations" rather than coupling per se, and just include coupling as one of the numerous possible mechanisms that could cause correlations to arise. The correlation between all-plant diversity and fungal diversity is of significant interest to conservation regardless of whether patterns are the result of direct "coupling", because vascular plant diversity is often used as a proxy for other hard-to-measure taxa like fungi when designing conservation priorities. Hopefully as more nuanced plant diversity geospatial layers become available that allow diversity of host plants to be extracted, more direct tests of diversity coupling can be performed to allow better understanding of the role of coupling in driving correlation patterns (see lines 327-332).

Minor comments

Figure 1B. The text in lines 35-37 conflates plant diversity and plant prevalence. Then the graph and the text in line 38 switches from host diversity to host prevalence, which sounds like cover/biomass instead of diversity. I think the coupled diversity argument is much stronger given the niche space expansion angle you cite. Regardless, please modify the sentence in lines 35-37 to separate these ideas.

Response: We have updated the text in the Introduction to explain more clearly that when talking about high host prevalence, we are talking about ecosystems where host plants comprise a larger proportion of the plant community. In such situations, vascular plant diversity is a stronger proxy for host diversity, and therefore we expect correlations between vascular plant and mycorrhizal fungal diversity to be stronger if diversity "coupling" is driving correlation structures. We have also updated the caption in Figure 1 to explain

this more clearly.

Figure 2. It's confusing to have biomes labeled as numbers in figure 2B and 2E. Label with the biomes instead. Similarly, Figure S3 would be more interpretable with biome designations in the figure.

Response: We have changed the numbers to biome names in Figure 2. However, we have kept the numbers in Figure S4 (originally Figure S3) due to a lack of space to fit the names.

Figure 3C. Make sure the values on the axes have the same significant digits. It would also be nice to have these panels on the same scale on the x and y axes to facilitate comparisons. Similar supplemental figures should follow suit.

Response: We have updated the figures as suggested.

Figure 4. This figure is not resolved enough to see the grid cells that are shaded darker to indicated consensus among datasets.

Response: We have now used darker shading so that the overlap is more obvious.

Figure S2. It would be helpful here to report the adjusted R2 as well as the deviance explained.

Response: We have now reported the adjusted R2 as well.

Ln 216. Change form to from.

Response: Changed.